# Malocclusion, Dental Caries and Oral Health-Related Quality of Life: A Comparison between Adolescent School Children in Urban and Rural Regions in Peru

**DOI:** 10.3390/ijerph17062038

**Published:** 2020-03-19

**Authors:** Maria Cadenas de Llano-Pérula, Estela Ricse, Steffen Fieuws, Guy Willems, Maria Fernanda Orellana-Valvekens

**Affiliations:** 1Department of Oral Health Sciences—Orthodontics, KU Leuven and Dentistry, University Hospitals Leuven, 3000 Leuven, Belgium; guy.willems@uzleuven.be; 2School of Dentistry, Universidad Peruana Cayetano Heredia, Lima 15102, Peru; estela_dentist@hotmail.com; 3Interuniversity Institute for Biostatistics and statistical Bioinformatics, KU Leuven and University Hasselt, 3000 Leuven, Belgium; steffen.fieuws@kuleuven.be; 4Department of Orthodontics, Erasme Hospital, Université Libre de Bruxelles, 1070 Brussels, Belgium; maria.valvekens@erasme.ulb.ac.be

**Keywords:** community-based study, occlusion/orthodontics, occlusal indices, caries prevalence, Oral Health Related Quality of Life

## Abstract

Rural, isolated areas benefit less from caries prevention programs and access to treatment than urban areas, and, hence, differences in oral health can be expected. This study aims to assess the prevalence of caries and malocclusion in urban and rural areas of Peru and to compare them with patients’ oral health perception. A total of 1062 adolescents were examined in five schools of rural (Titicaca) and urban (Lima and Cuzco) areas in Peru. Decay Missing Filled Teeth’s Surfaces, the Index of Complexity, Outcome and Need and the Child Oral Health Impact Profile short form-19 (COHIP-SF 19) were used to assess caries, severity of malocclusion and Oral Health Quality of Life, respectively. Significant differences in the prevalence (*p* = 0.001) and degree of caries (*p* = 0.001) were found between regions. The prevalence of caries was the highest in Cuzco (97.65%), followed by Titicaca (88.81%) and Lima (76.42%). The severity of malocclusion was the lowest in Titicaca. There was a negative relation between malocclusion, caries and COHIP-SF 19, with no evidence of a difference between the regions. This suggests that the higher the prevalence of caries and the more severe the malocclusion, the poorer the perception of oral health. In our study, rural areas presented a lower severity of malocclusion than urban areas.

## 1. Introduction

Dental caries is the most common chronic disease, according to the World Health Organization (WHO) [1]. Studies show that the lowest and highest prevalence of caries can be found in Africa and Latin America, respectively [2]. 

Malocclusion is also a common oral disturbance that involves a malposition of the jaws and/or the teeth. It has been shown that caries and malocclusion can affect the patient’s self-esteem and social abilities [3,4]. These domains are contained in a wider concept called Oral Health Related Quality of Life (OHRQoL), which has attracted increasing interest in the dental field in recent years. Quality of life is defined by the WHO as ‘the individual’s perception of their position in life within the culture context and value system they live in, considering their goals, expectations, standards, and concerns’ [5]. OHRQoL is also defined by the United Kingdom’s Department of Health as “a standard of health of the oral and related tissues which enables an individual to eat, speak and socialize without active disease, discomfort or embarrassment and which contributes to general well-being” [6]. 

In urban areas, access to dental, care and prevention programs should, in theory, decrease the prevalence of caries. In addition, the lack of exposure of children and adolescents from rural areas to factors such as television or social media raises the question of whether these elements affect their perception of their oral health. 

This project will address a series of fundamental gaps in our knowledge in this area. For example, little is known about the relationship between clinical measures of dental caries, malocclusion and OHRQoL. While there are standard clinical approaches for measuring dental caries and malocclusion, it is not known if the degree of malocclusion and dental caries, as determined by dental professional measures, has any relation with the perceived impact on the child. Several studies can be found regarding the prevalence of caries and malocclusion in specific areas in Latin America [7]. However, there is a paucity of studies comparing rural and urban regions within any given country.

Thus, the aim of this study was two-fold: (1) to assess the prevalence of caries and malocclusion and (2) to compare malocclusion, self-perception and caries between urban and rural areas in Peru by using the modified Index of Complexity Outcome and Need (ICON), the Child Oral Health Impact Profile short form-19 (COHIP-SF 19) and the Decay Missing Filled teeth’s Surfaces (DMFS) respectively.

## 2. Materials and Methods

This is a cross-sectional, multi-site, community-based, quantitative, epidemiological study. Since this was a Spanish-speaking population, written consent was obtained from the adolescents and their parents/caregivers as well as from all participants aged 18 and above in Spanish. The study and its voluntary nature were thoroughly explained by the native Spanish-speaking members of the team. Only those who gave consent were included in this study. The study was conducted in accordance with the Declaration of Helsinki, and it was approved by the Ethics Committee of the University Cayetano Heredia in Lima, Peru on the 7th of August 2015 with registration number 64478.

### 2.1. Selection Criteria

Two high schools in the cities of Lima, and one in Cuzco, and the isles of Taquille and Amantine, in the Lake Titicaca area were visited (five schools in total). Classes within the schools were sampled systematically. All students between 14 and 20 years of age attending school that day who had never undergone previous orthodontic treatment were examined and invited to participate.

OHRQoL was assessed by the short version of COHIP-SF 19 [8] in Spanish that consisted of 19 items (17 negative, two positive) forming five conceptually distinct domains (oral health, functional well-being, social-emotional well-being, school environment, and self-image) [9]. This questionnaire has been validated for a teenage population and was designed to be completed in 10–15 min. Questions were clarified by the investigators. For the two positive items, ordinal responses were recorded as “never = 0,” “almost never = 1,” “sometimes = 2,” “fairly often = 3”, and “almost all of the time = 4.” Scoring for the 17 negatively worded items was reversed. Higher COHIP-SF 19 scores reflected more positive OHRQoL, while lower scores reflected lower OHRQoL.

The dental examination comprised an extra oral assessment of the student’s smile and an intraoral examination of the teeth and occlusion. The DMFS index [10] was used to assess the dental caries experience and index of complexity outcome and need (ICON index) [11] to assess malocclusion. The examination lasted approximately 15 min per child, following the World Health Organization (2013) guidelines [12]. No radiographs, photos or study casts, were used. Personal data and information about orthodontic treatment were obtained directly from the students. Treatment urgency was dental examiner-determined by the status of oral disease present at the time of dental exam. Treatment urgency was scored by the dental examiner as: (1) See a dentist immediately, (2) See a dentist within two weeks, (3) See a dentist at earliest convenience, (4) Continue with routine care. This information was given to participants for their benefit, however, these data were not recorded. Each participant was given a toothbrush and oral hygiene instructions. The family was informed about their child’s oral health status. 

Clinical examination was carried out by three examiners, who had previously undergone calibration to standardize their procedures. Calibration exercises for raters of DMFS and ICON were performed using 10 casts and photos from patients that were not participating in the study. Intra and inter-rater reliability was assessed using 10 additional casts and photos from patients that were not part of the calibration exercises. These records were used to test each examiner at least three times, at 3–4 weeks apart between retests. 

The two locations in the Titicaca region represented isolated rural populations and the other two locations (Cuzco and Lima) corresponded to more developed urban areas where access to oral care and prevention programs was expected. In Lima, both a private and a public school were visited.

### 2.2. Statistical Methodology

Kruskall–Wallis tests, followed by pairwise Mann–Whitney U tests, and Chi^2^ tests, were used to compare between regions ordinal/continuous and categorical variables, respectively. Comparisons between regions were performed using a multivariable logistic regression model (for the prevalence of caries, as defined on the caries score), multivariable negative binomial models for count data (for the DMFS index and the mean number of surfaces with caries) and multivariable linear regression models (for total ICON score, the total COHIP-SF 19 score and the COHIP-SF 19 sub scores). In each of these models, age and gender were added as main effects. 

To compare the relation between the severity of the malocclusion (ICON score) and the Child Oral Health between regions, the multivariable linear regression models for the total COHIP-SF 19 score and the COHIP-SF 19 sub scores were extended with the ICON score as predictor, and with all two-way and three-way interactions. In univariable analyses, (Spearman) correlations between the ICON score and Child Oral Health were compared using a Z-test after applying a Fisher’s transformation. The same approach was used for the relation between the caries score and Child Oral Health. 

### 2.3. Based on These Models, the Following Questions Were Answered:

(1) Does the relation between ICON score and child oral health (measured by COHIP-SF 19) differ between regions? (2) Does this relation differ between males and females? (3) Does this relation depend on age? (4) Does the dependence of this relation on region differ between males and females? (5) Does the dependence of this relation on age differ between regions?

Note that questions 1–3 and questions 4–5 refer to two-way and three-way interactions, respectively. The caries score was right-skewed, and the assumption of linearity was—based on graphical exploration—not fulfilled in all settings. Therefore, analyses were repeated using a transformed version of the caries score which can handle the presence of zero values (inverse hyperbolic sign transformation, Burbridge et al. 1988 [13]). Since conclusions remained the same, results were only reported on untransformed values.

*p*-Values smaller than 0.05 were considered significant. No corrections for multiple testing were considered. Therefore, a single ‘significant’ *p*-value should be interpreted carefully. All analyses were performed using SAS software (SAS Institue Inc., North Carolina, NC, USA), version 9.4 of the SAS System for Windows. 

## 3. Results

Of the 1210 students who were invited to participate in this study, 1062 fulfilled the inclusion criteria; two participants were excluded from the analysis due to missing data. A total of 549 female (51.69%) and 513 male (48.31%) students with a mean age of 14.2 ± 1.67, participated in the study. See Table 1 for the demographic distribution of the sample.

Significant differences in the prevalence and degree of caries (as quantified by the caries score) were found between the regions, both when corrected and not corrected for differences in age distribution between the regions. Prevalence and degree of caries were the highest in Cuzco (97.65%), followed by Titicaca (88.81%) and then Lima (76.42%) (See Table 2 and Table 3). 

The severity of malocclusion (total ICON score) and the Subject’s Self-Evaluation of the Index of Orthodontic Treatment Need (IOTN) were significantly lower in Titicaca compared to Cuzco and Lima and (See Table 4 and Table 5). There was also a significant difference in Child Oral Health Perception between the regions. However, the direction of the differences depends on the specific scale (See Table 6 and Table 7). 

The correlations of each COHIP domain score (and the total score) with the global health self-rating question were significant although rather weak (all correlations were smaller than 0.30, see Table 8). 

There was a negative relationship between the severity of the malocclusion and the Child Oral Health, as well as between the degree of caries and the Child Oral Health. This holds for the total COHIP-SF 19 as well as for (most of) the subscales. Results regarding the relation between the ICON and the different COHIP-SF 19 subdomains can be found in Table 9.

Only a significant difference between regions could be detected for Social Economic Well-being, but this difference did not hold after correction for age and gender. For the rest of domains, and the total COHIP-SF 19 score, no significant interactions were found. Therefore, there is no indication that the relation between the severity of the malocclusion (ICON score) and the Child Oral Health differs between regions. The relation between the total COHIP-SF 19 and ICON scores per region can be visualized in Figure 1.

Regarding the relationship between the Caries score and the Child Oral Health, there is no strong evidence that this relationship differs between regions. Although there are some differences for some sub-scores (stronger negative correlations were found for Total COHIP-SF 19 score, Oral Health and school environment, for Cuzco vs. Titicaca, Cuzco vs. Lima and Cuzco + Lima vs. Titicaca, respectively), the pattern is not consistent (Table 10). 

Finally, there is a lack of evidence for moderating effects of gender and age. Only for some scales (COHIP-SF 19 total score and Oral Health) is there an indication that the relation becomes more negative at higher ages (detailed results not shown). 

## 4. Discussion

According to the findings of our study, the prevalence of caries in the studied populations in Peru is high, which is in agreement with previous articles [14]. In the total sample, 85.74% of the patients showed some degree of caries, of which 23.63% had pulpal involvement (Table 2). This can be explained by a lack of information regarding the importance of adequate oral hygiene. It has also been suggested that malnutrition and stunting can increase the occurrence of caries [14].

The novelty of our study lies in the comparison between rural and urban regions within one country. In this sense, it would have been interesting to record the diet and frequency of feeding habits of the included subjects, as one would expect differences in food consumption between rural, isolated areas and urban regions, which can have an impact on the occurrence of caries.

According to a recent study, a considerable percentage of Peruvian children do not brush their teeth, and this situation is more frequent in children living in rural areas [15]. Similar trends have also been detected in urban and rural areas of Chile, with a higher prevalence of caries in rural areas and, more significantly, in adolescents of 12 years old than in adolescents of 15 years old [16,17]. However, in our study, the prevalence of caries was not higher in rural areas. The highest prevalence was seen in Cuzco, followed by Titicaca and then Lima. Interestingly, pulpal involvement was the highest in the rural area of Titicaca, followed by Cuzco and then Lima, the biggest city, where it was significantly lower than in the other two locations. This could be due to a higher exposure to interceptive dental treatment in urban areas compared to rural, avoiding the evolution of carious lesions.

Additionally, we found the severity of malocclusion to be lower in Titicaca than in Cuzco and Lima. The ethnic background of this population may play a role in this. All population groups were Latinos with different origins, some of them indigenous (Andean, Amazonian, Quechuas) and, in the case of the Titicaca region, some coming from very isolated populations where the reduced number of inhabitants often leads to endogamy and a small gene pool [18]. In a recent case-control study on 15-year-olds living in urban and rural areas of India, a higher prevalence of malocclusion was also found in urban regions [17].

The Index of Complexity, Outcome and Need is a validated tool to determine the severity of malocclusion in a given patient. It takes several occlusal traits into consideration as well as an aesthetic assessment. However, this index does not perform well in mixed dentition, as some of the occlusal traits cannot be recorded correctly before full, definitive dentition has erupted. This somewhat conditioned the selection of our sample. The age range of our population is 14 to 20 years, which is normally when adolescents are orthodontically treated with fixed appliances. It could be argued that collecting information from younger subjects could have affected the results of the study, since orthodontic treatment involving maxillary expansion or functional therapy commonly starts at an earlier age, around 10–11 years.

Other indexes could have been used to determine malocclusion, such as the Index of Outcome and Treatment Need (IOTN) [19] or the Peer Assessment Record index (PAR) [20]. However, although they are both reliable indexes, it has been pointed out that they show a number of disadvantages when compared to ICON. For example, categorization using the Dental Health Component and the Aesthetic Component can be contradictory, with one suggesting treatment and the other not. They also have been validated against UK opinion and may not work for other countries, while the ICON has been validated in other countries [21,22].

When it comes to the perception of oral health, measured by the patient using the COHIP-SF 19 index, some significant differences seem to arise among regions, but they are connected to the specific subdomain. For example, in the domain of oral health, scores were the highest in Titicaca, while self-image seems to be better in Lima, then Cuzco and then Titicaca. The mean COHIP-SF 19 score was the highest in Lima, holding a significant difference with Cuzco (Table 5).

In the present investigation, a negative correlation between malocclusion, caries and OHRQoL was also found, meaning that the higher the prevalence of caries and the more severe the malocclusion, the lower the COHIP-SF 19 scores, and therefore the worse the reflected poorer perception of oral health by the patients, which is consistent with previously published studies [23]. However, in spite of this, no differences have been found in this relationship between regions, suggesting that it stays similar regardless of the area.

## 5. Conclusions

A total of 1062 adolescent schoolchildren from three different regions in Peru were examined to determine whether the prevalence of caries, the severity of malocclusion and the OHRQoL differed among rural (Lake Titicaca region) and urban areas (Cuzco, Lima). Results show a high prevalence of caries in Peru (85.74% of the included patients had some degree of caries), being the highest in Cuzco, followed by Titicaca and then Lima. The high prevalence of caries in Peru could be explained by a lack of information regarding oral hygiene. The severity of malocclusion and the Subject’s Self-Evaluation on IOTN were significantly lower in Titicaca compared to Cuzco and Lima. There is a negative relation between the severity of the malocclusion and the Child Oral Health, as well as between the degree of caries and the Child Oral Health. However, there is no indication that the relation between the severity of the malocclusion (ICON score) and the Child Oral Health differs between regions.

## Figures and Tables

**Figure 1 ijerph-17-02038-f001:**
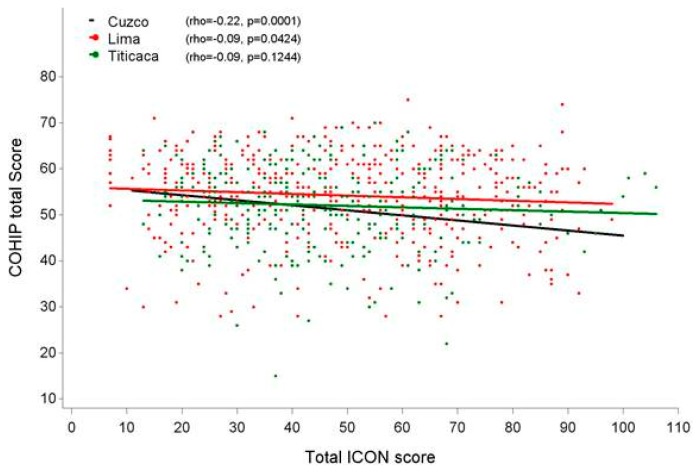
Relation between the ICON score and Child Oral Health amongst the regions.

**Table 1 ijerph-17-02038-t001:** Demographic distribution of the sample.

Variable	Statistic	Cuzco	Lima	Titicaca	Total	*p*-Value	Pairwise Comparisons		
							C vs. L	C vs. T	L vs. T
age	N	299	475	286	1060	<0.001	<0.001	0.014	0.030
	Mean	14.2	13.5	13.8	13.8				
	Std	1.67	1.56	1.92	1.72				
	Median	14.0	13.0	14.0	14.0				
	IQR	(13.0; 15.0)	(12.0; 15.0)	(12.0; 15.0)	(13.0; 15.0)				
	Range	(11.0; 20.0)	(11.0; 18.0)	(8.0; 19.0)	(8.0; 20.0)				
sex									
Female	*n*/*N* (%)	175/301 (58.14%)	240/475 (50.53%)	134/286 (46.85%)	549/1062 (51.69%)	0.019	0.038	0.006	0.326
Male	*n*/*N* (%)	126/301 (41.86%)	235/475 (49.47%)	152/286 (53.15%)	513/1062 (48.31%)				

Abbreviations: C: Cuzco, L: Lima, T: Titicaca, OR: odds ratio. CI: 95% confidence interval, IQR: Interquartile range, Std: Standard deviation.

**Table 2 ijerph-17-02038-t002:** Prevalence, degree of caries and Decay Missing Filled-Teeth’s Surfaces (DMFS) among the different regions.

Variable	Statistic	Cuzco	Lima	Titicaca	Total	*p*-Value	Pairwise Comparisons		
							C vs. L	C vs. T	L vs. T
Number of surfaces with caries	*N*	298	475	286	1059	<0.001	<0.001	0.001	<0.001
	Mean	7.6	4.3	6.2	5.7				
	Std	5.39	4.51	4.88	5.07				
	Median	7.0	3.0	6.0	5.0				
	IQR	(4.0; 9.0)	(1.0; 7.0)	(2.0; 9.0)	(2.0; 8.0)				
	Range	(0.0; 34.0)	(0.0; 21.0)	(0.0; 26.0)	(0.0; 34.0)				
Caries									
no caries	*n*/*N* (%)	7/298 (2.35%)	112/475 (23.58%)	32/286 (11.19%)	151/1059 (14.26%)	<0.001	<0.001	<0.001	<0.001
1–5 low	*n*/*N* (%)	101/298 (33.89%)	218/475 (45.89%)	107/286 (37.41%)	426/1059 (40.23%)				
6–10 moderate	*n*/*N* (%)	128/298 (42.95%)	99/475 (20.84%)	101/286 (35.31%)	328/1059 (30.97%)				
>10 severe	*n*/*N* (%)	62/298 (20.81%)	46/475 (9.68%)	46/286 (16.08%)	154/1059 (14.54%)				
Caries									
No caries	*n*/*N* (%)	7/298 (2.35%)	112/475 (23.58%)	32/286 (11.19%)	151/1059 (14.26%)	<0.001	<0.001	<0.001	<0.001
Caries	*n*/*N* (%)	291/298 (97.65%)	363/475 (76.42%)	254/286 (88.81%)	908/1059 (85.74%)				
DMFS	*N*	298	475	286	1059	<0.001	<0.001	<0.001	<0.001
	Mean	8.6	4.9	6.6	6.4				
	Std	5.49	4.76	4.91	5.25				
	Median	8.0	4.0	6.0	6.0				
	IQR	(5.0; 11.0)	(1.0; 7.0)	(3.0; 9.0)	(2.0; 9.0)				
	Range	(0.0; 34.0)	(0.0; 22.0)	(0.0; 26.0)	(0.0; 34.0)				
pulpal involvement									
No	*n*/*N* (%)	208/298 (69.80%)	421/474 (88.82%)	179/286 (62.59%)	808/1058 (76.37%)	<0.001	<0.001	0.065	<0.001
Yes	*n*/*N* (%)	90/298 (30.20%)	53/474 (11.18%)	107/286 (37.41%)	250/1058 (23.63%)				

**Table 3 ijerph-17-02038-t003:** Prevalence of caries, caries score and Decay Missing Filled-Teeth’s Surfaces (DMFS) index after correction for differences in age and gender distribution between the regions.

Caries Prevalence (% (CI))				Pairwise Differences (OR (CI))		
Cuzco	Lima		Titicaca	Cuzco vs. Lima	Cuzco vs. Titicaca	Lima vs. Titicaca
97.53% (94.90%; 98.82%)	77.56% (73.50%; 81.16%)		89.42% (85.31%; 92.48%)	11.40 (5.21; 24.93), *p* ≤ 0.0001	4.66 (2.01; 10.79), *p* = 0.0003	0.41 (0.27; 0.63), *p* ≤ 0.0001
**Mean Number of surfaces with caries (CI)**				**Pairwise ratios (CI)**		
**Cuzco**	**Lima**	**Titicaca**		**Cuzco vs. Lima**	**Cuzco vs. Titicaca**	**Lima vs. Titicaca**
7.26 (6.55; 8.04)	4.25 (3.91; 4.63)	6.17 (5.56; 6.86)		1.71 (1.49; 1.95), *p* ≤ 0.0001	1.18 (1.02; 1.36), *p* = 0.0301	0.69 (0.60;0.79), *p* ≤ 0.0001
**Mean DMFS (CI)**				**Pairwise ratios (CI)**		
**Cuzco**	**Lima**	**Titicaca**		**Cuzco vs. Lima**	**Cuzco vs. Titicaca**	**Lima vs. Titicaca**
8.23 (7.49; 9.05)	4.88 (4.51; 5.28)	6.55 (5.94; 7.22)		1.69 (1.49;1.91), *p* ≤ 0.0001	1.26 (1.10;1.44), *p* = 0.0009	0.74 (0.66;0.84), *p* ≤ 0.0001

Least-squares estimates of the prevalence are obtained from a multivariable logistic regression model. Results multivariable negative binomial model for count data (they represent the mean in a population of mean age and consisting of an equal number of males and females).

**Table 4 ijerph-17-02038-t004:** Total Icon of Complexity Outcome and Need (ICON) score per region.

Variable	Statistic	Cuzco	Lima	Titicaca	Total	*p*-Value	Pairwise Comparisons		
							C vs. L	C vs. T	L vs. T
**Total ICON score**	*N*	295	468	286	1049	0.019	0.651	0.004	0.031
	Mean	50.2	49.3	46.3	48.7				
	Std	19.14	22.00	18.74	20.40				
	Median	49.0	50.0	44.0	47.0				
	IQR	(37.0; 64.0)	(31.5; 67.0)	(32.0; 58.0)	(34.0; 64.0)				
	Range	(11.0; 100.0)	(7.0; 98.0)	(13.0; 106.0)	(7.0; 106.0)				

**Table 5 ijerph-17-02038-t005:** Total ICON score per region after correction for differences in age and gender distribution between the regions.

Mean Total ICON Score (CI)			Pairwise Differences (CI)		
Cuzco	Lima	Titicaca	Cuzco vs. Lima	Cuzco vs. Titicaca	Lima vs. Titicaca
50.80 (48.44; 53.15)	49.02 (47.18; 50.87)	46.27 (43.92; 48.63)	1.77 (−1.24; 4.78), *p* = 0.2482	4.52 (1.97; 7.86), *p* = 0.0079	2.75 (−0.24; 5.75), *p* = 0.0718

Results of the multivariable linear regression model. Mean Total ICON score in each region, after correction for age and gender There is a significant difference between the three regions (*p* = 0.0270).

**Table 6 ijerph-17-02038-t006:** Results of the Subject’s Self-Evaluation on Index of Orthodontic Treatment Need (IOTN).

Variable	Statistic	Cuzco	Lima	Titicaca	Total	*p*-Value	Pairwise Comparisons		
							1 vs. 2	1 vs. 3	2 vs. 3
Subjects Self Evaluation on IOTN	N	298	473	286	1057	<0.001	0.915	<0.001	<0.001
	Mean	3.1	3.0	2.4	2.9				
	Std	1.84	1.66	1.80	1.78				
	Median	3.0	3.0	2.0	3.0				
	IQR	(2.0; 4.0)	(2.0; 4.0)	(1.0; 3.0)	(2.0; 4.0)				
	Range	(1.0; 10.0)	(1.0; 10.0)	(1.0; 10.0)	(1.0; 10.0)				

**Table 7 ijerph-17-02038-t007:** Results of the Child Oral Health Impact Short-Form 19 (COHIP-SF 19) index per subdomain and region after correction for differences in age and gender distribution between the regions.

	Cuzco	Lima	Titicaca	Pairwise Differences (CI)		
				Cuzco vs. Lima	Cuzco vs. Titicaca	Lima vs. Titicaca
Mean COHIP-SF 19 Oral Health (CI)	5.11 (4.97; 5.25)	5.35 (5.24; 5.46)	5.62 (5.48; 5.77)	−0.24 (−0.42; −0.06), *p* = 0.0102	−0.51 (−0.71; −0.31), *p* ≤ 0.0001	−0.28 (−0.46; −0.10), *p* = 0.0028
Mean COHIP-SF 19 Functional Well Being (CI)	11.30 (10.99; 11.61)	11.77 (11.52; 12.01)	11.37 (11.06; 11.68)	−0.47 (−0.86; −0.07), *p* = 0.0201	−0.07 (−0.51; 0.37), *p* = 0.7557	0.40 (0.00; 0.79), *p* = 0.0479
Mean COHIP-SF 19 Social Emotional Well Being (CI)	16.50 (16.02; 16.98)	17.29 (16.92; 17.67)	16.14 (15.66; 16.63)	−0.79 (−1.40; −0.18), *p* = 0.0116	0.36 (−0.32; 1.04), *p* = 0.2988	1.15 (0.54; 1.76), *p* = 0.0002
Mean COHIP-SF 19 School Environment COHIP (CI)	6.07 (5.90; 6.24)	6.48 (6.35; 6.62)	6.51 (6.34; 6.68)	−0.41 (−0.63; −0.19), *p* = 0.0003	−0.44 (−0.68; −0.19), *p* = 0.0005	−0.03 (−0.25; 0.20), *p* = 0.8179
Mean COHIP-SF 19 Self Image (CI)	4.54 (4.32; 4.76)	5.30 (5.12; 5.47)	3.87 (3.65; 4.10)	−0.75 (−1.04; −0.47), *p* ≤ 0.0001	0.67 (0.35; 0.98), *p* ≤ 0.0001	1.42 (1.14; 1.70), *p* ≤ 0.0001
Mean COHIP-SF 19 total Score (CI)	51.21 (50.18; 52.23)	54.23 (53.42; 55.04)	51.96 (50.92; 52.99)	−3.02 (−4.34; −1.70), *p* ≤ 0.0001	−0.75 (−2.21; 0.72), *p* = 0.3160	2.27 (0.95; 3.59), *p* = 0.0007

Results of the multivariable linear regression model. Mean COHIP-SF 19 Self Image in each region, after correction for age and gender.

**Table 8 ijerph-17-02038-t008:** Relation between COHIP-SF 19′s different sub scores and Global Health self-rating question.

Global Health Self-Rating Question							
Region	“In General, You Think That Your Oral Health Is”:						
Frequency Row Pct	Poor	Fair	Average		Good	Excellent	Total
Cuzco	18	23	207		48	4	300
	6.00	7.67	69.00		16.00	1.33	
Lima	11	23	244		172	24	474
	2.32	4.85	51.48		36.29	5.06	
Titicaca	13	28	152		73	20	286
	4.55	9.79	53.15		25.52	6.99	
Total	42	74	603		293	48	1060
		**Spearman (95%CI)**					***p*-Value**
COHIP-SF 19 Oral Health		0.204		(0.146; 0.261)			<0.0001
COHIP-SF 19 Functional Well Being		0.199		(0.140; 0.256)			<0.0001
COHIP-SF 19 Social Emotional Well Being		0.221		(0.162; 0.277)			<0.0001
COHIP-SF 19 School Environment		0.125		(0.066; 0.184)			<0.0001
COHIP-SF 19 Self Image		0.194		(0.135; 0.251)			<0.0001
COHIP-SF 19 total Score		0.296		(0.240; 0.350)			<0.0001

*p*-value: raw *p*-value Spearman Correlation, based on Fishers Z transformation.

**Table 9 ijerph-17-02038-t009:** Relationship between the ICON and the different COHIP-SF 19 subdomains: Does the relationship depend on region?

	Total Score		Oral Health		Functional Well-Being		Social Emotional Well-Being		School Environment		Self-Image	
	Estimate (SE)	*p*-Value	Estimate (SE)	*p*-Value	Estimate (SE)	*p*-Value	Estimate (SE)	*p*-Value	Estimate (SE)	*p*-Value	Estimate (SE)	*p*-Value
Average slope	−0.059 (0.014)	<0.001	−0.009 (0.002)	<0.001	−0.006 (0.004)	0.1924	−0.022 (0.007)	0.001	0.002 (0.002)	0.4416	−0.009 (0.003)	0.0058
Does the relation depend on region?		0.0582		0.1385		0.6817		0.0302		0.2188		0.7947
-slope in Cuzco	−0.109 (0.027)	<0.001	−0.015 (0.004)	<0.001	−0.009 (0.008)	0.2566	−0.048 (0.013)	<0.001	−0.001 (0.005)	0.8052	−0.012 (0.006)	0.0453
-slope in Lima	−0.037 (0.019)	0.0511	−0.007 (0.003)	0.0048	−0.001 (0.006)	0.8443	−0.016 (0.009)	0.0769	0.007 (0.003)	0.0235	−0.008 (0.004)	0.0593
-slope in Titicaca	−0.031 (0.028)	0.2682	−0.006 (0.004)	0.1217	−0.007 (0.008)	0.4332	−0.003 (0.013)	0.8351	−0.000 (0.005)	0.9315	−0.006 (0.006)	0.2843

**Table 10 ijerph-17-02038-t010:** Relation between the Caries score and the different COHIP-SF 19 subdomains.

	Total Score		Oral Health		Functional Well-Being		Social Emotional Well-Being		School Environment		Self-Image	
	Estimate (SE)	*p*-Value	Estimate (SE)	*p*-Value	Estimate (SE)	*p*-Value	Estimate (SE)	*p*-Value	Estimate (SE)	*p*-Value	Estimate (SE)	*p*-Value
Average slope	−0.191 (0.057)	<0.001	−0.019 (0.008)	0.016	−0.067 (0.017)	<0.001	−0.059 (0.027)	0.0266	−0.031 (0.010)	0.0014	0.014 (0.012)	0.2496
Does the relation depend on region?		0.1028		0.2150		0.1100		0.2686		0.0353		0.2888
-slope in Cuzco	−0.359 (0.096)	<0.001	−0.036 (0.013)	0.0067	−0.115 (0.029)	<0.001	−0.115 (0.045)	0.0104	−0.049 (0.016)	0.0025	0.015 (0.021)	0.4802
-slope in Lima	−0.144 (0.091)	0.1137	−0.004 (0.013)	0.7452	−0.058 (0.027)	0.0323	−0.017 (0.042)	0.6938	−0.049 (0.015)	0.0013	−0.010 (0.020)	0.6120
-slope in Titicaca	−0.070 (0.108)	0.5178	−0.017 (0.015)	0.2608	−0.026 (0.032)	0.4149	−0.046 (0.051)	0.3694	−0.006 (0.018)	0.7336	0.038 (0.023)	0.1060

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
