# Peer review of "Malocclusion, Dental Caries and Oral Health-Related Quality of Life: A Comparison between Adolescent School Children in Urban and Rural Regions in Peru"

_ijerph, 2020, doi:10.3390/ijerph17062038_

Round 1
Reviewer 1 Report
This study aims the comparison of prevalence of malocclusion, dental caries and oral health-related quality of life (OHRQoL) between urban and rural regions.
Authors surveyed adolescents in rural (Titicaca) and urban (Lima and Cuzco) regions. ICON and CoHIP_SF were used to assess the severity of malocclusion and OHRQoL. However, some parts of descriptions and conclusions were different from other parts and contents of tables.
- The number of samples is not small. Authors used the nonparameteric statistics for significance tests. The description of the use of nonparameteric statistics instead of parameteric statistics.
- In abstract, page 1, line 22-23, Prevalence (dental caries) was the highest in Cuzco (97.65%), followed by Titicaca (88.81%), and Lima (76.42%). Table 2.b. also showed the same results. Cuzco and Lima are urban regions, and Titicaca is a rural region. However, in page 1, line 26-27 of abstracts, authors suggested “Rural areas present higher prevalence of caries and lower severity of malocclusion than urban areas. This conclusion was not based on the study results on caries prevalence.
- In results, page 4, line 142-143, authors described “The severity of malocclusion (total ICON score) and the Subject’s Self-Evaluation on IOTN were significantly lower in Titicaca compared to Cuzco and Lima and (See Tables 3a and 3b). However, in page 9, line 221-222 of discussion, ”Additionally, we found the severity of malocclusion to be higher in Titicaca than in Cuzco and Lima. The ethnical background of this populations may play a role in this.
But In page 1, line 26-27 of abstracts, authors suggested “Rural areas present higher prevalence of caries and lower severity of malocclusion than urban areas.
The severity of malocclusion of Titicaca as rural region was differently described.
- In table 2. a. the prevalence of Titicaca was the second (88.81%), lower than Cuzco (97.65%). However, The pulpal involvement of Titicaca in table 2. a. showed the highest rate (37.47%). The discussion should be needed the highest rate (37.47%) of pulpal involvement of Titicaca.
- A full unabbreviated term should be ahead of abbreviated term.
5.1. In abstracts, DMFS, ICON and COHIP-SF should be described as unabbreviated term for the understanding of readers.
5.2. In results, page 4, line 142, ‘IOTN’ should be described as ‘unabbreviated term (IOTN)’
5.3. In page 4, table 1 and page 6 table 2 3c, among pairwise comparisons, “1 vs 2, 1 vs 3, 2 vs 3” were not understandable for the indication of arabic number.
5.4. In page 4, table 2a and page 5 Table 2 3a among pairwise comparisons, “C vs L, C vs T, L vs T” should be noted as unabbreviated terms at the bottom of table 1.
5.5. In page 4, table 1, the unabbreviated term of IQR should be noted at the bottom of table 1.
5.6. In page 6 Table 2, 3b, table 2 3c, and page 7, table 4 and figure 1, the term of ‘ICON’ and ‘ITON’ should be noted as unabbreviated terms at the bottom of tables
Reviewer 2 Report
The study reads well and brings interesting insights towards the perception of oral health and prevalence of caries.
I believe the authors should further discuss the results with malocclusion, as they do not seem to make sense. A possible ilation is that malocclusion, within the age under study, is probably related to issues beyond perception of oral health. Treatment of malocclusion is generally implemented to individuals older than the ones being studied, therefore affecting less the perception of oral health.
Please provide more elements on this issue.
Reviewer 3 Report
The oral health of children and adolescents is one of the WHO's objectives, so this study provides data on the oral health conditions of adolescents in urban and rural areas in Peru. I consider that the methodologies were corrected to describe the results and this study could be useful in the future. The writing and organization of the article were well done. Also, I consider that this work is innovative and adds new information about this topic.
Round 2
Reviewer 1 Report
This study reported the comparison of prevalence of malocclusion, dental caries and oral health-related quality of life (OHRQoL) between urban and rural regions. The revised version included the enough answers on the reviewer's opinions. Authors employed enough samples and used the appropreate statistics for the analysis of data. This article is evaluated to contribute to desiging strategy of the oral health promotion for the students in remote region of shortage of oral health resources.